# Magnushammer: A Transformer-Based Approach to Premise Selection

**Maciej Mikuła**[*]
Google DeepMind[†]

**Szymon Tworkowski**[*]
xAI[†]

**Szymon Antoniak**[*]
Mistral AI[†]

**Bartosz Piotrowski**
IDEAS NCBR

**Albert Qiaochu Jiang**
University of Cambridge

**Jin Peng Zhou**
Cornell University[‡]

**Christian Szegedy**
xAI[‡]

**Łukasz Kuciński**
IDEAS NCBR,
IMPAN

**Piotr Miłoś**
IDEAS NCBR,
IMPAN

**Yuhuai Wu**
xAI[‡]

## Abstract

This paper presents a novel approach to premise selection, a crucial reasoning task in automated theorem proving. Traditionally, symbolic methods that rely on extensive domain knowledge and engineering effort are applied to this task. In contrast, this work demonstrates that contrastive training with the transformer architecture can achieve higher-quality retrieval of relevant premises, without the engineering overhead. Our method, Magnushammer, outperforms the most advanced and widely used automation tool in interactive theorem proving called Sledgehammer. On the PISA and miniF2F benchmarks Magnushammer achieves $59.5\%$ (against $38.3\%$) and $34.0\%$ (against $20.9\%$) success rates, respectively. By combining Magnushammer with a language-model-based automated theorem prover, we further improve the state-of-the-art proof success rate from $57.0\%$ to $71.0\%$ on the PISA benchmark using 4x fewer parameters. Moreover, we develop and open source a novel dataset for premise selection, containing textual representations of (*proof state*, *relevant premise*) pairs. To the best of our knowledge, this is the largest available premise selection dataset, and the first one for the Isabelle proof assistant.

## 1 Introduction

Automating mathematical reasoning has been a central theme of artificial intelligence since its earliest days (De Bruijn, 1970). Recently, machine learning has led to significant advancements in both informal (Lewkowycz et al., 2022) and formal mathematical reasoning (Kaliszyk and Urban, 2015b; Alemi et al., 2016; Polu and Sutskever, 2020; Han et al., 2022). The latter approach, adopted in this paper, allows mechanical verification of proofs by proof assistants.

Modern mathematics development is gradual: it feeds upon a huge body of already established knowledge and constantly adds to it. Proving a mathematical statement requires retrieval of facts from the knowledge base that can advance the proof. In automated reasoning literature, this retrieval process is known as *premise selection*.

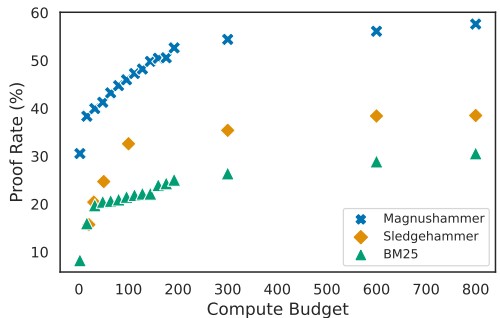

Figure 1: Proof success rate for varying computational budget for Magnushammer, Sledgehammer, and BM25. Magnushammer shows remarkable scalability. See Sections 5.1 for the definition of computational budget and Section 5.2.1 for configurations depicted in this figure.

---

[*]Equal contribution.

[†]Work performed while at the University of Warsaw.

[‡]Work performed while at Google Research.

Many tools have been developed to tackle premise selection (Alama et al., 2011; Kühlwein et al., 2012; Kaliszyk et al., 2017; Bansal et al., 2019), including a broad class known as "hammers," which leverage powerful automated theorem provers (ATPs) to determine useful premises (Paulson and Blanchette, 2012; Gauthier and Kaliszyk, 2015; Kaliszyk and Urban, 2015a; Czajka and Kaliszyk, 2018). One such tool, Sledgehammer (SH) (Paulson and Blanchette, 2012), has gained prominence with Isabelle (Paulson, 1993), where it helped to create a significant portion of Isabelle's proof corpus. Hammers are not yet available in all proof assistants (Ebner, 2020): implementing them is challenging due to the complex techniques required for different logics and type systems. There is a need for an effective premise selection tool that requires less adaptation to work for different proof assistants.

In this study, we provide a generic, data-driven, transformer-based (Vaswani et al., 2017) premise selection tool: Magnushammer. It constitutes a novel way to tackle the premise selection task, effective while requiring little domain-specific knowledge. Magnushammer is trained contrastively to perform premise retrieval in two stages: in the SELECT stage, it retrieves the most relevant $1024$ premises (measured by the cosine similarity of their embeddings to that of the current proof state) from tens of thousands (the database contains 433K premises in total and typically 30K–50K are available in each proof state); in the RERANK stage, the retrieved premises are re-ranked with proof-state-aware scores: tokens of the proof state directly attend to tokens of the premise, giving a more contextualized relevance score. An overview of Magnushammer's architecture is shown in Figure 2b.

Magnushammer can prove $59.5\%$ of the theorems on the PISA benchmark (Jiang et al., 2021), a substantial improvement over Sledgehammer's $38.3\%$. We demonstrate that this dominance is consistent with varying controlled compute budgets, shown in Figure 1. Furthermore, we replace the premise selection component (Sledgehammer) in a neural-symbolic model Thor (Jiang et al., 2022a) with Magnushammer and improve the state-of-the-art proof success rate on PISA from $57\%$ to $71\%$.

To train Magnushammer, we extracted a premise selection dataset from the Isabelle theorem prover and its human proof libraries. The dataset consists of $4.4$M premise selection instances, with 433K unique premises. To the best of our knowledge, this is the largest open-sourced premise selection dataset, and the first one of this kind for Isabelle. We find Magnushammer to be data efficient, outperforming Sledgehammer with only 4K training examples ($0.1\%$ of the training data available). The main contributions of this work are the following:

- We propose the use of transformers trained contrastively as a novel way of addressing the premise selection problem. Our method, Magnushammer, achieves a $59.5\%$ proof rate on the PISA benchmark, significantly improving the $38.3\%$ proof rate of Sledgehammer, the most powerful general-purpose automation tool for Isabelle.
- We extract and open source the largest, to the best of our knowledge, premise selection dataset. It consists of $4.4$M premise selection examples and 433K unique premises.
- We analyze how Magnushammer's performance depends on the model size, dataset size, and the inference-time compute budget. We show its superiority with moderate resources.

## 2 BACKGROUND: PROOF ASSISTANTS, ISABELLE, AND SLEDGEHAMMER

Proof assistants (aka interactive theorem provers, or ITPs) such as Isabelle (Paulson, 1993), Lean (de Moura et al., 2015), Coq (Bertot, 2008), HOL Light (Harrison, 1996), or Mizar (Grabowski et al., 2010), are software tools designed to assist the development of formal proofs. They provide expressive language for the formalization of mathematical statements and proofs while verifying them formally. In Isabelle, theorems are proved sequentially: an initial *proof state* is obtained after the theorem is stated, and the proof state changes when the user provides a valid *proof step* (see Appendix A.1 for an example). Proof states contain information about the already established facts and the remaining goals to prove. Proof steps consist of *tactics*, which are optionally parametrized by *premises*. Tactics are theorem-proving procedures and can complete some proofs in one step provided with relevant premises. However, finding these premises is difficult: one needs to select a handful of relevant facts from the current proof context, which typically contains tens of thousands of them.

Sledgehammer (Paulson and Blanchette, 2012; Blanchette et al., 2013) is a powerful automated reasoning tool for Isabelle. It belongs to a broader class of tools known as "hammers," which integrate automated theorem provers (ATPs) into proof assistants. The goal of these tools is to support the process of finding and applying proof methods. Sledgehammer has become an indispensable tool for Isabelle practitioners (Paulson and Blanchette, 2012). It allows for closing low-level gaps between subsequent high-level steps of proof without the need to memorize entire lemma libraries.

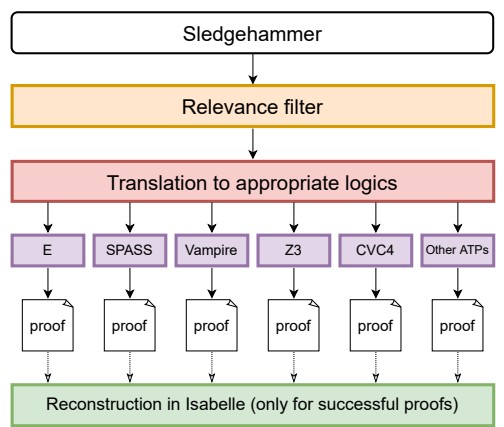

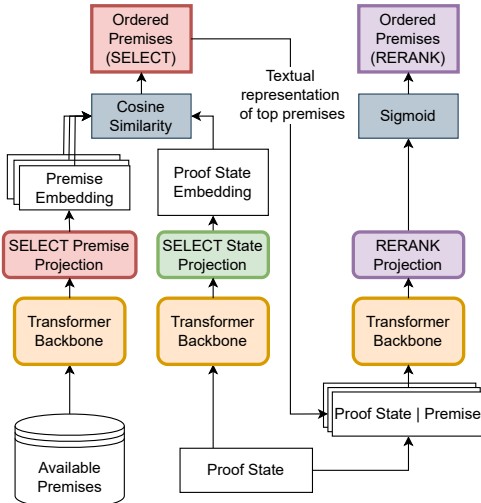

(a) A call to Sledgehammer triggers the following sequence of steps: First, available facts are filtered based on their similarity to the conjecture. Then, the conjecture together with the selected facts (usually a few hundred in number) are translated to simpler logic used by the external provers (E, SPASS, etc.). Then, such problems are fed into each ATP separately. Finally, the premises used in the successful ATP proofs are used to reconstruct a proof inside Isabelle using its native methods.

(b) Given a proof state, we first retrieve the most relevant premises according to the cosine similarity of their embeddings with the proof state embedding (SELECT). We then re-rank these with a model that encodes each proof state and premise pair, outputting a relevance score (RERANK). The bulk of the architecture is a shared transformer model, in orange.

Figure 2: Overview of Sledgehammer (a) and Magnushammer (b).

Sledgehammer is designed to first pre-select a number of relevant facts heuristically, translate them together with a conjecture to simpler logic, and try to prove the conjecture using strong, external ATPs like E (Schulz, 2004), SPASS (Weidenbach, 2001), Vampire (Kovács and Voronkov, 2013), Z3 (de Moura and Bjørner, 2008), or cvc5 (Barbosa et al., 2022). If successful, these provers generate complete proofs. They are, however, not trusted by Isabelle. Instead, the facts used in the external proofs are extracted and used to produce a proof *inside* Isabelle using its native methods. Up to this last step, known as *proof reconstruction*, Sledgehammer is essentially used as a precise premise selection tool. See Figure 2a depicting the whole process.

While immensely useful, Sledgehammer comes with several limitations. First, increasing computational power for Sledgehammer brings quickly diminishing returns (Böhme and Nipkow, 2010). Second, the logic projection and proof reconstruction in a hammer are not straightforward for type systems other than higher-order logic (Czajka and Kaliszyk, 2018). Finally, Sledgehammer's performance hinges on the relevance filtering scheme, a suite of methods based on handcrafted heuristics (Meng and Paulson, 2009) or classical machine learning (Kühlwein et al., 2013). Such approaches are unlikely to efficiently utilize the constantly growing body of proof data.

We argue that all these limitations can be overcome with deep-learning-based approaches. Neural networks have shown remarkable effectiveness in end-to-end problem solving with little or no feature engineering (Krizhevsky et al., 2012; Brown et al., 2020). Adopting textual representations with generic neural solutions removes the need for logic projection, ATP solving, and proof reconstruction. Moreover, large language models have recently displayed impressive scaling properties with respect to both model size (Kaplan et al., 2020) and data (Hoffmann et al., 2022).

## 3 MAGNUSHAMMER

The goal of premise selection is to find relevant mathematical facts for a given proof state. We focus on selecting premises with a neural model informed by their textual representations instead of relying on fact structures like Sledgehammer (see Section 2). The core idea of Magnushammer is to combine fast retrieval based on representational similarity (SELECT) with a more accurate re-ranking (RERANK), as outlined in Algorithm 1. Our method closely follows those of Nogueira and Cho (2019) and Izacard et al. (2021). This hierarchical approach is scalable to large formal libraries containing hundreds of thousands of facts. Below we describe the two-stage Magnushammer approach.

SELECT leverages *representation similarity* and is based on batch-contrastive learning similar to the methods of Alemi et al. (2016), Bansal et al. (2019), Han et al. (2021), or Radford et al. (2021). SELECT embeds premises and proof states into a common latent space and uses cosine similarity to determine their relevance. During inference, it requires only one pass of a neural network to compute the proof state embedding and dot product with cached premise embeddings. SELECT is hence fast and scalable to large sets of premises. In our experiments, there are between 30K and 50K premises in a typical proof state context, from which we select $K_S = 1024$ most relevant ones.

RERANK scores the relevance of the $K_S$ selected premises for the current proof state by analyzing the (`proof_state`, `premise`) pairs. RERANK is trained to output the probability of the `premise` being relevant to the `proof_state`. The $K_S$ premises retrieved by SELECT are re-ranked with respect to these probabilities, and the final list comprises of the top $K_R$ premises (we set $K_R = K_S$). Having both the premise and the proof state in a single input allows RERANK to be more accurate. However, at the same time, it is much slower, as each pair must be scored individually.

---

**Algorithm 1** Premise selection with Magnushammer.

---

**Require:**
    `proof_state`, `premises`         ▷ proof state to retrieve premises for and database of available premises
    $K_S, K_R$                  ▷ number of premises to retrieve with SELECT and RERANK, respectively
1: `state_embedding` ← `get_embeddings(proof_state)`           ▷ SELECT stage starts
2: `premises_embeddings` ← `get_embeddings(premises)`
3: `Cache(premises_embeddings)`
4: `sim_scores` = `state_embedding` · `premises_embeddings`
5: `selected` = `premises[argsort(−sim_scores)[: `$K_S$`]]`
6: `batch` = `[]`                        ▷ RERANK stage starts
7: **for** `premise` in `selected` **do**
8:     `batch.append((proof_state, premise))`
9: `rerank_scores` ← `get_rerank_scores(batch)`
10: `top_premises` = `selected[argsort(−rerank_scores)[: `$K_R$`]]`
11: **return** `top_premises`

---

**Training** We train Magnushammer using two alternating tasks: SELECT is trained with a modified InfoNCE loss (van den Oord et al., 2018), and RERANK is trained with the standard binary cross-entropy loss. The architecture of Magnushammer shares a transformer backbone with specialized linear projections on top (see Figure 2b). The backbone is pre-trained with a language modeling task on the GitHub and arXiv subsets of the Pile dataset (Gao et al., 2021). For training, we use datasets consisting of (`proof_state`, `premise`) pairs extracted with a procedure described in Section 4.

During SELECT's training, each batch consists of $N$ proof states, $N$ positive premises (one for each proof state), and additional $M$ negative premises sampled from available facts that are not ground truth premises for any of the selected proof states. This gives $N − 1 + M$ negatives per proof state in one batch. We typically use $M = 3N$, which differs from standard batch-contrastive learning (Radford et al., 2021), in which $M = 0$ and negatives are only the other $N − 1$ premises in the batch RERANK is trained using a binary classification objective. For each positive (`proof_state`, `premise`) pair in the dataset, we construct 15 negatives from the most likely false positives returned by SELECT. Specifically, all the premises $\mathcal{M}$ that are facts that were never used as a premise for `proof_state`, are first chosen. Then, the top 1024 of $\mathcal{M}$ according to SELECT are selected, and 15 are sampled from them to construct negative training pairs. See Appendix B for complete training details.

**Evaluation in Isabelle** We outline how premises chosen by Magnushammer are used to prove theorems in Isabelle. Given a proof state, a list of the $k$ most relevant premises $P$ is retrieved. We construct proof steps consisting of a tactic $t$ and a subset of premises $S \subseteq P$. Such proof steps are executed in parallel, with a timeout of 2 seconds. The evaluation is successful if any of these proof steps completes the proof. For $S$, we pick the top $i$ of $P$, where $i$'s are consecutive powers of 2 up to $2^{10}$, or 0 for tactics that do not accept premises. More details, including the set of tactics used, are presented in Appendix D. An example of a proof with tactics and premises is given in Appendix A.3.

Note that the procedure of trying multiple different subsets of premises is commonly applied in the context of automated theorem proving (Urban et al., 2008; Kühlwein et al., 2012) and similar to the technique implemented in Sledgehammer (Paulson and Blanchette, 2012). The rationale behind this is that the proof procedures implemented in ATPs and high-level ITPs' tactics perform combinatorial search, and providing them with fewer premises to restrict their search space is beneficial.

## 4 DATASETS

We created and released[1] a comprehensive dataset of textual representations for Isabelle's proof states and premises.To the best of our knowledge, this is the first high-quality dataset of this kind for Isabelle, and also the largest premise selection dataset overall. We used the two largest collections of Isabelle theories to create the dataset: the Archive of Formal Proofs and the Isabelle Standard library.

For every proof step in every proof from these collections, we extracted the preceding proof state and the set of premises used in the proof step; this was turned into $(\texttt{proof\_state}, \texttt{premise})$ pairs constituting training data points. We call this the HUMAN PROOFS LIBRARY (HPL) dataset. In addition, we used Sledgehammer to generate proofs that are different from the human ones by using potentially alternative premises. We refer to this as the SH partition, and its union with HPL constitutes the MACHINE-AUGMENTED PROOFS LIBRARY (MAPL) dataset. Statistics for all these datasets are given in Table 1. Note that MAPL grosses over 4M data points.

Below we describe in more detail how data points are extracted from a proof step. An Isabelle's proof is a sequence of $(\texttt{proof\_state}, \texttt{proof\_step})$ pairs: $\texttt{proof\_state}$ has the state information, and $\texttt{proof\_step}$ is a tactic application that advances the proof. A $\texttt{proof\_step}$ may use $\texttt{premises}$: theorems, lemmas, or definitions established previously. Suppose a $\texttt{proof\_step}$ contains $n$ premises: $p_1, p_2, \ldots, p_n$. We then extract $n$ data points: $(\texttt{proof\_state}, p_1), \ldots, (\texttt{proof\_state}, p_n)$. Executing Sledgehammer on the $\texttt{proof\_state}$ may result in multiple different synthetic $\texttt{proof\_steps}$, and data points can be extracted from each in the same way (see Appendix A.2 for details).

Table 1: Statistics of MAPL and both its partitions: HPL (coming from human-written proofs) and SH (coming from Sledgehammer-generated proofs). The data points are of the form of $(\texttt{proof\_state}, \texttt{premise})$ pairs.

| Dataset | HPL | SH | MAPL |
|---|---|---|---|
| Data points | 1.1M | 3.3M | 4.4M |
| Unique proof states | 570K | 500K | 570K |
| Unique premises | 300K | 306K | 433K |

Mining the HPL partition took 10K CPU hours, and mining the SH partition took 150K CPU hours (17 CPU years) on a distributed system.

Our datasets have 2 distinguishing features:

1. The human-originating dataset is augmented by alternatives generated with Sledgehammer, which results in a significantly larger and more diverse dataset. This also decreases the probability of sampling *false negatives* while training contrastively: a negative example $(\texttt{proof\_state}, \texttt{premise})$ may in fact be positive, but we just have not seen an alternative proof using $\texttt{premise}$. Generating multiple alternative proofs partially remedies this problem.
2. Both $\texttt{proof\_states}$ and $\texttt{premises}$ are represented as "high-level" Isabelle's text instead of "low-level" logical formalism like, e.g., TPTP (Sutcliffe, 2017) used by Alama et al. (2014). This makes the dataset more suitable for language models, decreases the need for feature engineering, and facilitates cross-proof-assistant pre-training (Conneau and Lample, 2019).

## 5 EXPERIMENTS

We evaluate Magnushammer on the PISA and miniF2F theorem proving benchmarks using *proof success rate* as a metric. Our main result is that Magnushammer outperforms Sledgehammer by a large margin and, combined with Thor (Jiang et al., 2022a), sets a new state of the art on the PISA benchmark (71.0% from 57.0%). Through ablations, we study the effectiveness of Magnushammer and the contribution of its components. Additional results and details can be found in Appendix E.

### 5.1 EXPERIMENTAL DETAILS

**Benchmarks** For evaluation, we use PISA (Jiang et al., 2021) and miniF2F (Zheng et al., 2022) benchmarks. PISA contains problems randomly selected from the Archive of Formal Proofs;[2] we use the same 1000 problems as Jiang et al. (2022a) for our evaluations. miniF2F consists of 488 high-school competition-level problems, split into validation and test set, each with 244 problems.

---

[1]`https://huggingface.co/datasets/Simontwice/premise_selection_in_isabelle`
[2]When training on data from the Archive of Formal Proofs, we remove the subset of it appearing in PISA.

Table 2: Proof rates on the PISA benchmark. On the single-step task, Magnushammer outperforms both Sledgehammer and BM25 by a wide margin. On the multi-step task, Magnushammer combined with Thor achieves the state-of-the-art proof rate of $71.0\%$.

| Task | Method | Proof rate (%) |
|---|---|---|
| Single-step | BM25 | 30.6 |
| | TF-IDF | 31.8 |
| | OpenAI embed. (Neelakantan et al., 2022) | 36.1 |
| | Sledgehammer | 38.3 |
| | Magnushammer (ours) | **59.5** |
| Multi-step | LISA (Jiang et al., 2021) | 33.2 |
| | Thor (Jiang et al., 2022a) | 57.0 |
| | Thor + Magnushammer (ours) | **71.0** |

Table 3: Proof rates on the miniF2F benchmark. On the single-step task, Magnushammer outperforms Sledgehammer and its variant with additional heuristics (Jiang et al., 2022b). On the multi-step task, Thor+Magnushammer obtains competitive results, significantly outperforming Thor+Sledgehammer.

| Task | Method | Valid (%) | Test (%) |
|---|---|---|---|
| Single-step | Sledgehammer | 9.9 | 10.4 |
| | Sledgehammer + heuristics | 18.0 | 20.9 |
| | Magnushammer (ours) | **33.6** | **34.0** |
| Multi-step | Thor + Sledgehammer (Jiang et al., 2022a) | 28.3 | 29.9 |
| | Thor + Sledgehammer + auto (Wu et al., 2022a) | 37.3 | 35.2 |
| | Thor + Magnushammer (ours) | 36.9 | 37.3 |
| | DSP (Jiang et al., 2022b) | **43.9** | **39.3** |

**Metric and evaluation setups** To evaluate the performance, we measure *proof success rate*: the percentage of successful proofs. A proof is successful if it is formally verified by Isabelle. We distinguish *single-step* and *multi-step* settings. In the single-step setting, we check if the theorem can be proven in one step by applying premises retrieved by the evaluated premise selection method (e.g., Magnushammer). In the multi-step scenario, we perform a proof search using a language model following Thor (Jiang et al., 2022a). Thor + Magnushammer uses Magnushammer instead of Sledgehammer as the premise selection component. A further explanation is given in Section 5.2.

**Evaluation protocol and computational budget** Algorithm 3 (in Appendix D) details the evaluation of Magnushammer in the single-step setting. It generates $|\mathcal{T}| \times |K|$ proof steps by combining each tactic $t \in \mathcal{T}$ with top $k$ premises from a ranking provided by Magnushammer, where $\mathcal{T}$ is a prescribed set of tactics, $k \in K$, and $K$ is a list of integers. Such constructed proof steps are then executed in Isabelle. We define the computational budget for such an evaluation as $C = |\mathcal{T}| \times |K| \times T$, where $T$ is a timeout expressed in seconds (we use $T = 2$ s as we observed little benefit from increasing it). Estimating the computational budget for Sledgehammer is difficult due to its complex internal architecture. We approximate it by $C = S \times T$, where $S$ is the 'number of CPU cores' (corresponding to steps executed in parallel) and $T$ is the timeout. We use $S = 10$ for our calculations. See Appendix A.4 for more details.

**Architecture and training details** For our main experiments, we pre-train standard decoder-only transformer models with 38M and 86M non-embedding parameters and fine-tune them for downstream tasks of premise selection or proof step generation. Full details are given in Appendix C. In our experiments, we use the Portal-to-ISAbelle API (Jiang et al., 2021) to interact with Isabelle.

## 5.2 RESULTS ON PISA AND MINIF2F BENCHMARKS

Our main empirical results, summarized in Table 2 and Table 3, were obtained with the 86M parameter model. Figure 1 and Section 5.2.1 deepen this study, showing that Magnushammer outperforms Sledgehammer across a broad spectrum of computational budgets.

**Performance on the single-step task** In the single-step setting, Magnushammer outperforms Sledgehammer by a wide margin on both PISA ($59.5\%$ vs. $38.3\%$) and miniF2F ($34.0\%$ vs. $20.9\%$). Additionally, on PISA, Magnushammer outperforms TF-IDF and BM25: text-based, non-trainable

retrieval methods (Robertson and Zaragoza, 2009) which are strong baselines in common retrieval benchmarks (Thakur et al., 2021). This suggests that Magnushammer is able to learn more than just superficial text similarity. In all these experiments we used the same evaluation protocol (following Algorithm 3) and computational budget of 1000 as detailed in Appendix D.1.

Interestingly, retrieval based on the generic OpenAI embeddings (Neelakantan et al., 2022) (specifically: text-embedding-ada-002) yields reasonable performance comparable to Sledgehammer. This confirms the potential of neural premise selection to replace traditional symbolic methods. There is, however, a large gap to match Magnushammer. This shows that contrastive fine-tuning on our dataset provides non-trivial gains and supports our hypothesis that Magnushammer learns more than just mere textual similarity exploited by the general purpose method.

**Performance on the multi-step task** Neural theorem provers utilize language models to generate proof steps, following the approach proposed by Polu and Sutskever (2020). This allows for the creation of more complex, multi-step proofs. The proof generation involves sampling a proof step from the language model, verifying it, and repeating this process until the proof is closed or the computational budget is exceeded. The best-first search algorithm is often used to explore the most promising proof steps.

Thor (Jiang et al., 2022a) augments neural theorem provers with premise-selection capabilities. To this end, Thor allows the model to generate proof steps using Sledgehammer, which we replace with Magnushammer (see Appendix D.2 for details). Thor + Magnushammer establishes a new state of the art on the PISA benchmark ($71.0\%$ vs. $57.0\%$). On miniF2F, our method also significantly outperforms Thor and achieves results competitive with the current state of the art. In these experiments, we give Magnushammer a computational budget of 200.

It is important to note that other theorem-proving approaches in the multi-step section of Table 3 require much larger language models: for Thor it is 700M non-embedding parameters; DSP (Draft, Sketch, and Prove) by Jiang et al. (2022b) uses Minerva model (Lewkowycz et al., 2022) with 62B parameters. Moreover, these other approaches rely on ideas orthogonal to premise selection. Specifically, Thor + auto (Wu et al., 2022a) proposes a variation of Thor, involving expert iteration on auto-formalized data. DSP involves creating a high-level outline of a proof and uses Sledgehammer to solve the low-level subproblems. We hypothesize that both methods would perform even better when combined with Magnushammer.

### 5.2.1 SCALING COMPUTATIONAL BUDGET

In this section, we discuss how the quality of premise selection methods varies with the computational budget available during evaluation. Figure 1 shows the results, and the definition of the compute budget is provided in Section 5.1. Notably, Magnushammer outperforms Sledgehammer even with very limited computational resources, and it scales well, particularly within the medium budget range.

For Magnushammer and BM25, we use Algorithm 3 (Appendix D) in various configurations (i.e., settings of $\mathcal{T}$ and $K$). We start with one tactic, $\mathcal{T} = \{\texttt{smt}\}$, and $K = [2^7]$, which yields $C = 2$ (recall that $T = 2$ s). We then gradually add more tactics to $\mathcal{T}$ and more values to $K$. The final setup uses $|\mathcal{T}| = 36$ and $K$ containing all powers of 2, from $2^0$ up to $2^{10}$, which yields $C \approx 800$. The details are provided in Appendix D. For Sledgehammer, we scale the timeout parameter $T$ up to 80 s.

### 5.3 IMPACT OF TRAINING DATA

We study how the amount and type of data impact the proof success rate by comparing HPL and MAPL datasets. For this comparison, we used models with 38M non-embedding parameters and a computational budget of 800.

**Dataset size** Our method is data-efficient: see Figure 3a. We observe that Magnushammer fine-tuned on only $0.1\%$ of MAPL – equivalent to approximately 4K samples – is already able to outperform Sledgehammer. This indicates that when starting from a pre-trained model, Magnushammer is a promising approach for addressing premise selection in theorem-proving environments with limited training data. The effect of pre-training diminishes as the amount of training data increases.

**Dataset type** Fine-tuning on MAPL or HPL leads to subtle differences ($56.3\%$ vs. $54.0\%$ when the whole datasets are used). This outcome may be attributed to the impact of model pre-training and the fact that the HPL dataset is rich enough to obtain good performance on the PISA benchmark (as observed in the previous paragraph). We speculate that the bigger MAPL dataset might be essential for future harder benchmarks and scaling up the model size.

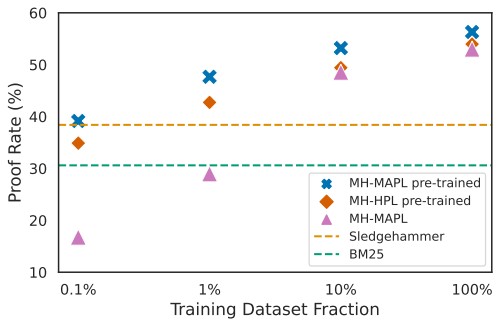 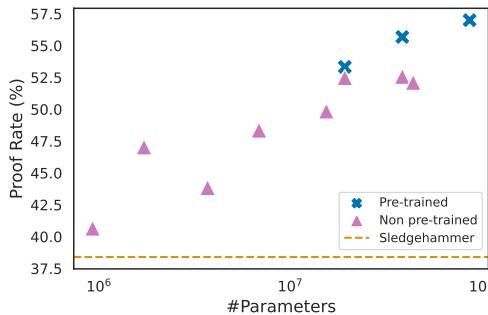

(a) We randomly sample fractions of MAPL or HPL datasets and use them for training Magnushammer. Even 0.1% of the MAPL dataset allows pre-trained Magnushammer to outperform the Sledgehammer and BM25 baselines. See Table 4 for numerical data.

(b) We train Magnushammer of different sizes. Even with a one-layer transformer, Magnushammer outperforms Sledgehammer. We observe consistent performance gains with increasing model sizes. Pre-trained models perform better. See Table 5 for numerical data.

Figure 3: Impacts of the training data quantity and the model parameters on the proof rate. The vertical axis is the proof rate in percentage. In Subfigure 3a, the horizontal axis is the fraction of training dataset used and in Subfigure 3b it is the number of parameters in the model.

## 5.4 ABLATIONS

We use models trained on the MAPL dataset and evaluate them with a computational budget of 800. To study how the performance of our method depends on the model size, we vary the number of layers $L$ and embedding dimension $D$. A positive correlation between the model size and the proof rate is shown in Figure 3b. We observe that even a tiny model with 920K parameters ($L = 1, D = 256$) outperforms Sledgehammer (40.7% vs. 38.3%). We also note the benefit of pre-training and that scaling the number of layers is more beneficial than scaling the embedding dimension. The details can be found in Appendix C.1. The impact of re-ranking is studied in Appendix C.5.

## 6 RELATED WORK

Premise selection becomes a crucial task whenever proving theorems automatically within a large formal library. Moreover, this task has several unique aspects that are challenging from the perspective of learning-based approaches. Therefore, there exist multiple works that tackle learning premise selection (either explicitly or implicitly) applying various methods focusing on different aspects.

Many works employ classical machine learning like Bayesian and kernel methods (Kühlwein et al., 2012; Alama et al., 2014), $k$-NN (Blanchette et al., 2016), or decision trees (Piotrowski and Urban, 2018; Nagashima and He, 2018; Piotrowski et al., 2023). The common weakness of these approaches is the necessity of using hand-engineered features, whereas faster, simpler training is an advantage.

Alemi et al. (2016) were the first to apply deep learning to premise selection, thus dispensing with the hand-designed features completely. Their approach was evaluated in an automated theorem proving setting and not in a proof assistant, as is Magnushammer. They also implicitly learn embeddings of conjectures and premises, which are concatenated and passed through a shallow network, whereas the training signal comes from the logistic loss. In contrast, Magnushammer demonstrated the strength of training with the contrastive loss, where the obtained embeddings just need to be passed through a simple cosine similarity measure to provide high-quality rankings.

Most of the methods explicitly targeting the premise selection problem (including this work) retrieve a *ranking* of independently treated premises. In contrast, Piotrowski and Urban (2020) aimed at modelling the implicit dependencies *between* the premises and used LSTM-based language models to produce structured sequences of premises. However, the premises were treated there as opaque tokens, not giving the neural model the ability to inspect the statements of the premises.

Effective deep learning approaches often leverage the explicit structure of mathematical expressions using graph neural networks (Wang et al., 2017; Paliwal et al., 2020; Goertzel et al., 2022). Our work uses the transformer architecture (Vaswani et al., 2017), which is highly scalable and capable of producing powerful representations of raw text data.

Pre-trained transformer language models have been applied to various aspects of theorem proving, including autoformalization (Wu et al., 2022a; Jiang et al., 2022b), conjecturing (Urban and Jakubuv, 2020), and tactic prediction / proof step search (Yang and Deng, 2019; Polu and Sutskever, 2020; Han et al., 2022; Lample et al., 2022; Polu et al., 2023). The works from the last category often implicitly deal with premise selection by treating premises as names / tokens to be generated and not inspecting their statements. The application of generative language models to statement-aware premise selection has been limited, as the length of the possible premises often greatly exceeds the context of several thousand tokens that the models are designed to handle. Thor (Jiang et al., 2022a) circumvents the difficulty of premise selection by invoking Sledgehammer. In contrast, Magnushammer *retrieves* rather than *generates* to overcome the context length limitation. Therefore it can be used in tandem with other models (its combination with Thor is demonstrated in Section 5).

Batch-contrastive learning is widely used in speech (van den Oord et al., 2018), text (Izacard et al., 2021), image (Chen et al., 2020) and image-text (Radford et al., 2021) representation learning. These methods have proven effective despite the possibility of false negatives occurring in contrastive batches (Robinson et al., 2021). The SELECT phase of our premise selection model relies on in-batch negative examples to train the retriever, similar to HOList (Bansal et al., 2019) and Contriever (Izacard et al., 2021). Like HOList, we mine additional negatives, which we found crucial for performance. The RERANK stage closely resembles (Nogueira and Cho, 2019), but instead of using BM25, we jointly train retrieval and re-ranking, utilizing premises retrieved by SELECT as hard negatives for RERANK training. Han et al. (2021) use contrastive learning in informal premise selection. Concurrently to our work, Yang et al. (2023) develop a premise selection method for Lean also using contrastive learning in a way similar to our SELECT method, but without the RERANK stage.

There are multiple lines of work considering datasets based on formal theorem proving. These include benchmarks like ProofNet (Azerbayev et al., 2022) for Lean, and miniF2F (Zheng et al., 2022) that supports multiple ITPs. These datasets only focus on evaluation, not providing data for training the models. Another line of research focuses on benchmarking machine learning models' reasoning capabilities while also providing training data (Bansal et al., 2019; Li et al., 2021; Han et al., 2022). Existing public datasets for premise selection include the ones introduced in (Alama et al., 2014; Piotrowski and Urban, 2020). In comparison to these works, we publish the data in high-level, textual format, as seen in Isabelle, instead of low-level, structured languages such as TPTP (Sutcliffe, 2017).

There exists a rich body of work developing complex hammers systems for different proof assistants (Paulson and Blanchette, 2012; Kaliszyk and Urban, 2015a; Gauthier and Kaliszyk, 2015; Czajka and Kaliszyk, 2018). Unlike the traditional hammers, our method does not depend on external ATPs and requires little domain-specific knowledge.

# 7 LIMITATIONS AND FUTURE WORK

**Other proof assistants** Magnushammer treats proof states and premises as text and makes no assumptions about their structure. As such, no feature engineering is needed to apply it to other proof assistants. We conjecture that Magnushammer can prove effective in other environments because it is agnostic to the logic or type system used. We plan to evaluate Magnushammer in Lean proof assistant on ProofNet (Azerbayev et al., 2022) and miniF2F (Zheng et al., 2022) benchmarks, using the recently published LeanDojo toolkit (Yang et al., 2023) that also provides baselines for comparison.

**Richer proof and premise representations** Magnushammer utilizes the textual representation of the proof state given by Isabelle. This representation, however, does not provide complete semantic information about the referenced objects. Including function definitions and object types in the proof state representation might further improve performance.

**Modelling full proof steps** Combining language models with external premise selection tools significantly improves their theorem-proving performance, as demonstrated by Jiang et al. (2022a) and our work. A natural step would be to further integrate premise selection with language models into a single model capable of generating proof steps containing relevant retrieved premises. A proof of concept of this idea was explored by Tworkowski et al. (2022). This would also allow to model existing implicit dependencies between returned premises, which was shown beneficial by Piotrowski and Urban (2020). We believe that recent advances in retrieval-augmented language models (Wu et al., 2022b; Borgeaud et al., 2022) could facilitate progress in this direction.

## ACKNOWLEDGEMENTS

We gratefully acknowledge that our research was supported with Cloud TPUs from Google's TPU Research Cloud (TRC). Moreover, Piotr Miłoś was supported by the Polish National Science Centre grant 2019/35/O/ST6/03464.

## REPRODUCIBILITY STATEMENT

The data that were used for pre-training of the backbone transformer model of Magnushammer are freely available under this link: `https://pile.eleuther.ai/`

The Isabelle data used in training for the down-stream tasks are available under this link: `https://huggingface.co/datasets/Simontwice/premise_selection_in_isabelle`

The benchmarks used for evaluation of Magnushammer are freely available on GitHub:

- miniF2F: `https://github.com/openai/miniF2F`
- PISA: `https://github.com/albertqjiang/Portal-to-ISAbelle`

PISA also implements the interface for interacting with Isabelle that we used in our experiments.

Appendix A.4 specifies the setup of Sledgehammer that we used in our comparisons. Appendices B and C detail the shape of the transformer architecture used, define the loss functions applied in the SELECT and RERANK stages, specify the hyperparameters used in pre-training and training for our down-stream tasks, and disclose the hardware used for training. Appendix D details the setup for evaluation of Magnushammer in Isabelle, in particular the list of tactics applied on top of the Magnushammer's premise selection.

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

APPENDIX

# A    ISABELLE ENVIRONMENT

This section contains visual examples of proofs in Isabelle and provides some configuration details of the environment.

## A.1    VISUALIZATION OF THE ISABELLE ENVIRONMENT

Figure A.1 shows an example theorem and its proof, as seen in Isabelle's most popular IDE, jEdit. The theorem comes from an entry to the Archive of Formal Proofs – *Fun With Functions* (Nipkow, 2008). It states that any mapping $f$ from the set of natural numbers to itself that satisfies $f(f(n)) < f(n+1)$ must be the identity function. The proof starts with a simple induction and then refines the result to arrive at the thesis. This problem was included in Terence Tao's booklet *Solving Mathematical Problems* (Tao, 2010).

```
theorem identity1: fixes f :: "nat ⇒ nat"
assumes fff: "⋀n. f(f(n)) < f(Suc(n))"
shows "f(n) = n"
proof -
  { fix m n have key: "n ≤ m ⟹ n ≤ f(m)"
    proof(induct n arbitrary: m)
      case 0 show ?case by simp
    next
      case (Suc n)
      hence "m ≠ 0" by simp
      then obtain k where [simp]: "m = Suc k" by (metis not0_implies_Suc)
      have "n ≤ f(k)" using Suc by simp
      hence "n ≤ f(f(k))" using Suc by simp
      also have "... < f(m)" using fff by simp
      finally show ?case by simp
    qed }
  hence "⋀n. n ≤ f(n)" by simp
  hence "⋀n. f(n) < f(Suc n)" by(metis fff order_le_less_trans)
  hence "f(n) < n+1" by (metis fff lift_Suc_mono_less_iff[of f] Suc_eq_plus1)
  with ‹n ≤ f(n)› show "f n = n" by arith
qed
```

Figure A.1: An example theorem in Isabelle. The statement is highlighted in the orange frame and the body of the proof is in the green frame. In this proof, most of the lines contain two consecutive steps: the first formulates a new proposition, and the second proves it. See a detailed analysis of the line 8 of the proof in Figure A.2 below.

```
proof (state)
this:
  m = Suc k
goal (1 subgoal):
  1. ⋀n m. (⋀m. n ≤ m ⟹ n ≤ f m)
     ⟹ Suc n ≤ m ⟹ Suc n ≤ f m
```

```
then obtain k where [simp]: "m = Suc k" by (metis not0_implies_Suc)
```

```
not0_implies_Suc:"n ≠ 0 ⟹ ∃m. n = Suc m"
```

Figure A.2: The line is broken down into two steps: the first one (green frame) includes the proposition (since $m$ is natural and positive, it must have a predecessor $k$) and the second (blue frame) proves it using the tactic `metis` with premise `not0_implies_Suc`, that states that a nonnegative natural number is a successor of some other natural number. The used premise is a fact which is already defined in the lemma library. The proof state resulting from the first step is in the yellow frame. The full premise statement is highlighted in pink.

## A.2 Alternative proof step generation with Sledgehammer

This section describes how to generate alternative proof steps using Sledgehammer which we do to obtain datasets described in Section 4. First, we find all intermediate propositions within the proof (they can be nested) and try to replace the proof of the proposition with a Sledgehammer step. If successful, we record such a step in the dataset and proceed with both the original and the alternative proof. Figure A.3 provides a visual example of the aforementioned propositions.

```
proof -
  { fix m n have key: "n ≤ m ⟹ n ≤ f(m)"
    proof(induct n arbitrary: m)
      case 0 show ?case by simp
    next
      case (Suc n)
      hence "m ≠ 0" by simp
      then obtain k where [simp]: "m = Suc k" by (metis not0_implies_Suc)
      have "n ≤ f(k)" using Suc by simp
      hence "n ≤ f(f(k))" using Suc by simp
      also have "... < f(m)" using fff by simp
      finally show ?case by simp
    qed }
  hence "⋀n. n ≤ f(n)" by simp
  hence "⋀n. f(n) < f(Suc n)" by(metis fff order_le_less_trans)
  hence "f(n) < n+1" by (metis fff lift_Suc_mono_less_iff[of f] Suc_eq_plus1)
  with ‹n ≤ f(n)› show "f n = n" by arith
```

Figure A.3: Example intermediate propositions highlighted in red. Note: not all propositions were highlighted.

## A.3 Example of a proof with tactics requiring premises

Figure A.4 contains a multi-step proof of the irrationality of $\sqrt{2}$ written in Isabelle. The proof contains multiple usages of tactics that require premises.

```
lemma "sqrt 2 ∉ ℚ"
proof
  assume "sqrt 2 ∈ ℚ"
  then obtain a b::int where "sqrt 2 = a/b" "coprime a b" "b ≠ 0"
    by (metis Rats_cases' less_irrefl)
  then have c: "2 = a^2 / b^2"
    by (smt (z3) of_int_power power_divide real_sqrt_pow2)
  then have "b^2 ≠ 0" by fastforce
  then have *: "2*b^2 = a^2"
    by (smt (verit, ccfv_SIG) c comm_semiring_class.distrib
        eq_divide_eq_numeral(1) mult_cancel_right1 numeral_Bit0
        numeral_plus_numeral of_int_add of_int_power
        of_int_power_eq_of_int_cancel_iff one_plus_numeral)
  then have "even a"
    by (smt (z3) even_power oddE)
  then obtain c::int where "a=2*c" by blast
  with * have "b^2 = 2*c^2" by auto
  then have "even b"
    by (smt (z3) even_power oddE)
  with ‹coprime a b› ‹even a› ‹even b› show False by fastforce
qed
```

Figure A.4: A proof of $\sqrt{2} \notin \mathbb{Q}$ (Jiang et al., 2022a, Figure 1). The steps containing `metis`, `smt`, `fastforce`, `blast`, `auto`, `fastforce` are examples of steps using premises. For instance, one such proof step is `by (metis Rats_cases' less_irrefl)`. This step invokes `metis` and provides two premises as arguments, namely `Rats_cases'` and `less_irrefl`.

## A.4 SLEDGEHAMMER SETUP

We set up Sledgehammer in Isabelle 2021-1, following the configuration used by Jiang et al. (2022a). We run Sledgehammer using different sets of settings and calculate the total proof rate by taking the union of problems solved by each run. The Sledgehammer timeout is set to default 30 seconds. We use only on-machine automated theorem provers (same as Isabelle environment), so external provers used by Sledgehammer are the following: Z3, SPASS, Vampire, CVC4, and E.

In our calculation of the Sledgehammer computation budget, see Section 5.1, we assume $S = 10$ 'CPU cores.' We run our experiments on machines with 96 CPU cores, making the assumption realistic. Moreover, we emphasize that the performance gap between Magnushammer and Sledgehammer is large enough that altering the value of $S$, e.g., to an unrealistic level $S = 1$, would not qualitatively change conclusions.

## B   DETAILS OF MAGNUSHAMMER

### B.1   SELECT STAGE

SELECT stage is trained using the InfoNCE loss van den Oord et al. (2018) defined as:

$$\mathcal{L}\left(q, k_{+}\right) = -\frac{\exp\left(s\left(q, k_{+}\right)/\tau\right)}{\exp\left(s\left(q, k_{+}\right)/\tau\right) + \sum_{i=1}^{K} \exp\left(s\left(q, k_{i}\right)/\tau\right)},$$

where $q$ is a query (a proof state), $k_{+}$ is a positive premise (a ground truth from the dataset), $k_i$ are negative premises. We define $s$ as cosine similarity between proof state and premise embeddings; $\tau > 0$ is a non-trainable temperature parameter. We list our hyperparameter choices in section C.2.

### B.2   RERANK STAGE

Premise retrieval task can be cast as binary classification, trying to determine if a given pair $(\texttt{proof\_state}, \texttt{premise})$ is relevant. Applying classification to each pair is computationally infeasible, however, it could be used to *re-rank* a small set of premises retrieved by SELECT. Namely, we use the following cross-entropy loss:

$$\mathcal{L} = -\sum_{p \in \mathcal{P}} \log \texttt{score}(p) - \sum_{p \notin \mathcal{N}} \log(1 - \texttt{score}(p)),$$

where $\texttt{score}(p)$ is the output of the RERANK part of the model (see "Sigmoid" in Figure 2b) for a given $p = (\texttt{proof\_state}, \texttt{premise})$ pair. Typically, we sample a batch of 16 positive pairs $\mathcal{P}$ from the dataset. For each such pair $(\texttt{proof\_state}, \texttt{premise})$ 15 negatives are constructed from the most likely false positives returned by SELECT. Specifically, negative premises $\mathcal{M}$, which are facts that were never used as a premise for $\texttt{proof\_state}$, are first chosen. Then, the top 1024 of $\mathcal{M}$ according to SELECT are selected, and 15 are sampled from them to construct negative pairs, which are included in $\mathcal{N}$.

### B.3   MAGNUSHAMMER

We train Magnushammer as two separate tasks alternating update steps as presented in Algorithm 2. Note that the backbone of the architecture is shared between SELECT and RERANK, thus such multi-task training is potentially more effective than having two separate models. Calculation of the negative premises for SELECT is costly, thus for efficiency reasons we recalculate the top 1024 premises, see Section B.2, every $T = 1000$ steps in the $\texttt{recompute\_negatives\_for\_rerank}$ function, as outlined in the Algorithm 2.

## C   TRAINING DETAILS

### C.1   MODEL ARCHITECTURE

We use a decoder-only transformer architecture, following the setup from Wang and Komatsuzaki (2021) and using rotary position embedding by Su et al. (2021), a variation of relative positional

---

**Algorithm 2** Magnushammer training.

---

**Require:**
    $\theta$                $\triangleright$ initial trainable parameters
    $D$                $\triangleright$ premise dataset
    $T$                $\triangleright$ interval for updating rerank dataset
 1: $D_{\mathrm{rerank}} \leftarrow$ `recompute_negatives_for_rerank`$(\theta, D)$
 2: $\mathrm{step} = 0$
 3: **while** $\mathrm{step} <$ `num_train_steps` **do**
 4:     `batch_select` $\leftarrow D.\mathrm{sample}()$
 5:     $\theta \leftarrow \mathrm{train\_step}(\theta, \texttt{batch\_select})$
 6:     `batch_rerank` $\leftarrow D_{\mathrm{rerank}}.\mathrm{sample}()$
 7:     $\theta \leftarrow \mathrm{train\_step}(\theta, \texttt{batch\_rerank})$
 8:     $\mathrm{step} \leftarrow \mathrm{step} + 1$
 9:     **if** $\mathrm{step} \mod T = 0$ **then**
10:         $D_{\mathrm{rerank}} \leftarrow$ `recompute_negatives_for_rerank`$(\theta, D)$

---

encoding. The feedforward dimension in the transformer block is set to $4 \times D$ where $D$ denotes embedding dimension, and the number of attention heads is $H = D/64$. Our 38M model has $L = 12$ layers and an embedding dimension of $D = 512$. The larger 86M model consists of $L = 12$ layers and has $D = 768$. For all the models, we use the original GPT-2 tokenizer (Radford et al., 2019).

In SELECT, we append a specialized token at the end of the sequence to compute the embedding for a proof state and linearly project its embedding. Premises are embedded analogously. Similarly to Radford et al. (2021) that train separate projections for images and captions, we train separate proof state and premise projections and share the transformer backbone (see Figure 2b). Analogously for RERANK, we compute the relevance score by taking the embedding of the last token and then projecting it to a scalar value.

## C.2 HYPERPARAMETER SETUP

We performed the following hyperparameter sweeps. We note that we have not observed significant differences between obtained results.

- Learning rate: $\{1e{-}4, 2e{-}4, 3e{-}4, 5e{-}4\}$, chosen: $2e{-}4$
- Dropout: $\{0.0, 0.05, 0.1, 0.2\}$, chosen: $0.1$
- Weight decay: $\{0.02, 0.05, 0.1\}$, chosen: $0.02$
- Batch size $N$ in SELECT: $\{128, 256, 512\}$, chosen: $256$
- Number of negatives $M$ in SELECT: $\{0, 256, 768, 1536\}$, chosen: $768$
- Temperature for InfoNCE loss in SELECT: $\{0.05, 0.07, 0.2, 1\}$, chosen: $0.07$
- Batch size for RERANK: $\{16, 32, 64\}$, chosen $64$
- Number of negatives per proof state $\mathcal{M}$ in RERANK: $\{7, 15\}$, chosen: $15$.

## C.3 PRE-TRAINING ON LANGUAGE MODELING

Pre-training has been shown to dramatically increase the capabilities and performance of decoder-only models on tasks other than language modeling (Howard and Ruder, 2018). Motivated by that, we pre-train our models on GitHub and arXiv subsets of the Pile (Gao et al., 2021). The models are trained for 1M steps, with a context length of 2048. Global batch size is set to 32 sequences giving a total number of 65536 tokens per batch. Dropout is disabled, and weight decay is set to 0.02. The learning rate increases linearly from 0 to 0.0003 for the first 10000 steps, and then the cosine schedule is applied to decrease its value gradually.

## C.4 FINE-TUNING FOR DOWNSTREAM TASKS

We train Magnushammer by taking a pre-trained language model, removing its language modeling head, and attaching three linear projections heads – one projection for proof state embedding, another

one for premise embedding, and the last one for producing relevance score for RERANK, as depicted in Figure 2b and described in Section C.1. For the proof step generation task, we fine-tune our language models by applying the algorithm used to train Thor (Jiang et al., 2022a).

## C.5 IMPACT OF RE-RANKING

We find that the SELECT-only method, i.e., Magnushammer without the RERANK phase, already significantly outperforms Sledgehammer. Tested on the 38M model, it achieves a $54.2\%$ proof rate comparable to $56.3\%$ obtained by Magnushammer. SELECT-only mode is a computationally appealing alternative, as it only needs a single forward pass to embed the current proof state (the setting used recently by Yang et al. (2023).) Premise embeddings can be pre-computed and cached, allowing inference on the CPU without the need for GPU or TPU accelerators.

## C.6 HARDWARE

We gratefully acknowledge that our research was supported with Cloud TPUs from Google's TPU Research Cloud (TRC). We use TPU virtual machines from the Google Cloud Platform (GCP) for all stages: pre-training, fine-tuning, and evaluation. Each TPU virtual machine has 8 TPU v3 cores, 96 CPU cores, and over 300GB of RAM. TPU v3 cores have around 16GB of memory each. The Isabelle environment is set to have access to 32 CPU cores.

## D MAGNUSHAMMER EVALUATION

In Algorithm 3 we outline our evaluation method described in Section 5.1. To generate proof steps, we use the following tactics: `smt`, `metis`, `auto`, `simp`, `blast`, `meson`, `force`, `eval`, `presburger`, `linarith`. Algorithm 3 is also used to evaluate BM25, where we select `top_premises` with this retrieval method instead of Magnushammer.

### D.1 COMPUTATIONAL BUDGET

For our main result (Section 5.2), we allocate the computational budget of 1000 as follows: apart from the powers of two from $2^0$ to $2^{10}$, we also try the following $k$ values: $[48, 96, 192]$, which in total gives 14 values. With each of these $k$ values, 36 tactics are used with timeout $T = 2$, yielding $C \approx 1000$.

For the ablation studies, we only use powers of two from $2^0$ to $2^{10}$, and the same set of 36 tactics, which gives $C \approx 800$.

---

**Algorithm 3** Magnushammer evaluation in ITP environment.

---

**Require:**
    theorem                                               ▷ theorem to prove
    premsel_model                     ▷ Magnushammer's premise selection model
    $K_S$                   ▷ number of premises to retrieve with SELECT
    $K_R$                  ▷ number of premises to retrieve with RERANK
    premises                          ▷ available premises
    top_k_premises_to_try     ▷ list with the number of top premises to generate steps with
    tactics_to_try                   ▷ list of tactics to generate steps with
    env                        ▷ ITP environment (e.g., Isabelle)
1: proof_state ← init_problem(env, theorem)         ▷ initialize problem
2: top_premises ← premsel_model(proof_state, premises, $K_S, K_R$)     ▷ get top premises
3: steps = []               ▷ generate proof steps combining of tactics and top $k$ premises
4: **for** k **in** top_k_premises_to_try **do**
5:     top_k_premises ← top_premises[: k]
6:     new_steps ← generate_steps(tactics_to_try, top_k_premises)
7:     steps.extend(new_steps)
8: solved ← try_steps(env, steps) ▷ evaluate generated proof steps in the ITP's environment
9: **return** solved

---

## D.2 THOR + MAGNUSHAMMER

To generate more complex proofs we combine Thor (Jiang et al., 2022a) with Magnushammer as introduced in multi-step setting in Section 5.2.

Firstly, we follow the procedure described in Jiang et al. (2022a) to pre-process training data and fine-tune our pre-trained language model for the proof generation task (pre-training details can be found in Appendix C.3). During the evaluation, when the language model generates the `<hammer>` token, we call our method instead of Sledgehammer. More specifically, we use an augmented Algorithm 3 that returns the proof states resulting from applying the steps (instead of returning binary information on whether any of the steps closed the proof). We then pick at most s = 2 states among these and add them to the BFS queue.

We assign the same computational budget as proposed in Thor, with the only difference being that each `proof_step` has a timeout limit of 2 s (instead of 10 s), which we found to perform better in our setup. The search is terminated if and only if one of the following scenarios happens: (1) a valid proof has been found for the theorem; (2) the language model is queried 300 times; (3) a wall-time timeout of 500 s has been reached (assuming parallel execution of Magnushammer steps); (4) the queue is empty but the theorem is not proved. We keep the same maximum length of the queue equal to 32.

## E  ADDITIONAL EXPERIMENTAL RESULTS

### E.1  SUPPLEMENTAL DETAILS

We provide additional details for our main experiments and ablations.

Table 4: Relation between the training data and the proof rate discussed in Section 5.3 and Figure 3a.

| Dataset | Fraction | Pre-trained | Proof rate (%) |
|---------|----------|-------------|----------------|
| MAPL    | 0.1%     | Yes         | 39.2           |
| HPL     | 0.1%     | Yes         | 34.9           |
| MAPL    | 0.1%     | No          | 16.8           |
| MAPL    | 1%       | Yes         | 47.7           |
| HPL     | 1%       | Yes         | 42.7           |
| MAPL    | 1%       | No          | 29.0           |
| MAPL    | 10%      | Yes         | 53.2           |
| HPL     | 10%      | Yes         | 49.4           |
| MAPL    | 10%      | No          | 48.5           |
| MAPL    | 100%     | Yes         | 56.3           |
| HPL     | 100%     | Yes         | 54.0           |
| MAPL    | 100%     | No          | 53.0           |

### E.2  STEP TACTIC PROMPT

We observed that different tactics use different subsets of premises. This motivated us to extend the context given to our model with *tactic prompt*. Namely, provide the tactic name as an additional argument to the premise selection model, similarly to Bansal et al. (2019). Prompting model with the tactic name does not yield significant improvements. However, it allows the model for a more accurate premise selection. Namely, as presented in Figure A.5 and Table 6, we observe that premises necessary to close the proof are ranked higher. This motivates an alternative performance metric presented in the next section.

Table 5: Proof rate on PISA for different models discussed in Section 5.4 and Figure 3b. We vary the number of layers $L$ and the embedding dimension $D$ of the Transformer model.

| Transformer $(L, D)$ | #Parameters | Pre-trained | Proof rate (%) |
|---|---|---|---|
| $(1, 256)$ | 920K | No | 40.7 |
| $(1, 512)$ | 3.7M | No | 43.9 |
| $(2, 256)$ | 1.7M | No | 47.0 |
| $(2, 512)$ | 6.8M | No | 48.4 |
| $(2, 768)$ | 15.4M | No | 49.9 |
| $(6, 512)$ | 19.2M | No | 52.5 |
| $(6, 512)$ | 19.2M | Yes | 53.3 |
| $(6, 768)$ | 43.7M | No | 52.1 |
| $(12, 512)$ | 38.3M | No | 52.6 |
| $(12, 512)$ | 38.3M | Yes | 56.3 |
| $(12, 768)$ | 86.2M | Yes | 57.0 |

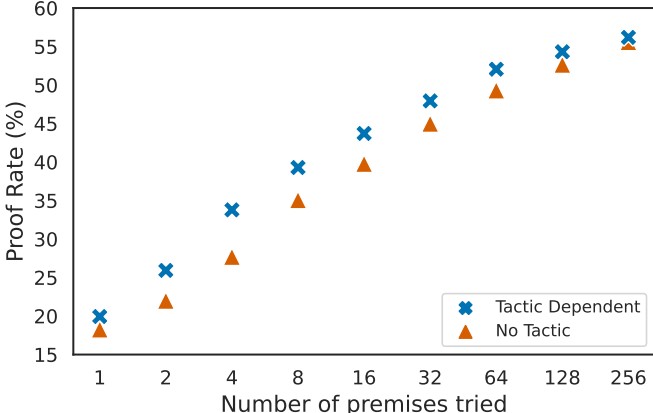

Figure A.5: We calculate *accumulated proof rate* in the following way: try 1 premise, count problems solved, then try 2 premises, count problems solved using 1 or 2 premises, then try 4 premises, count problems solved using 1, 2, or 4 premises etc. Following this, on the x-axis we have the number of premises used to generate steps in Algorithm 3. The y-axis presents the accumulative proof rate as we try more and more premises. The higher the proof rate for the smaller number of premises used the better. We observe that prompting the model with the tactic is not necessary to achieve the final high proof. However, it allows the model for a more accurate premise selection – all premises necessary to close the proof are ranked higher.

Table 6: Effect of the number of premises used for generating tactic steps on the proof rate. We fix a set of tactics and accumulate problems solved as we increase the number of premises used to generate steps in Algorithm 3. Namely, for each $k$, we count the number of problems solved using at most $k$ premises. The "Tactic" column indicates whether the model was given a tactic prompt.

| Model | Tactic | Dataset | $k \leq 0$ | $\leq 1$ | $\leq 2$ | $\leq 4$ | $\leq 8$ | $\leq 16$ | $\leq 32$ | $\leq 64$ | $\leq 128$ | $\leq 256$ |
|---|---|---|---|---|---|---|---|---|---|---|---|---|
| BM25 | No | N/A | 9.63 | 13.56 | 15.62 | 16.70 | 18.47 | 20.73 | 23.38 | 25.44 | 28.00 | 30.55 |
| MH-86M | No | HPL | 9.63 | 19.25 | 22.99 | 28.68 | 34.58 | 39.88 | 44.50 | 47.84 | 51.47 | 52.95 |
| MH-86M | Yes | HPL | 9.63 | 20.24 | 25.44 | 31.53 | 36.15 | 40.67 | 44.70 | 48.53 | 51.87 | 54.22 |
| MH-86M | No | MAPL | 9.63 | 18.27 | 22.00 | 27.70 | 35.07 | 39.78 | 44.99 | 49.31 | 52.65 | 55.60 |
| MH-86M | Yes | MAPL | 9.63 | 19.94 | 25.93 | 33.79 | 39.29 | 43.71 | 47.94 | 52.06 | 54.32 | 56.19 |

### E.3    NUMBER OF PREMISES USED AS A PERFORMANCE METRIC FOR PREMISE SELECTION.

Consider the number of premises used to generate steps in Algorithm 3 (parameter $k$ in the for-loop). Intuitively, the fewer premises needed the better, since it means that all the premises necessary to close the proof are ranked higher (high recall), thus the model does a more accurate premise selection. In other words, a better retrieval model should be able to score all the necessary facts higher and push unnecessary facts down the list.

To compare different models we fix a set of tactics and accumulate problems solved as we increase the number of premises used to generate steps in Algorithm 3. This is presented in Table 6 and Figure A.5. Namely, for each $k$, we count the number of problems solved using at most $k$ premises. Effectively, each new value of $k$ adds one new step per tactic to try.

### E.4    SINGLE-STEP PROOF RATE BOUND

It is non-trivial to estimate the lower bound on how many problems can be closed directly from the root state in a single proof step. To answer this question, we use different models in Algorithm 3 and take the union of problems solved by them. Namely, we ensemble the results of the Magnushammer variations introduced in previous sections: Magnushammer-86M, Magnushammer-38M, Magnushammer-SELECT, Sledgehammer, BM25, and the models presented in Section 5.4. Such a combination successfully closes $65.5\%$ of the proofs.

## F    EXAMPLES OF PROOFS FOUND BY MAGNUSHAMMER

In Sledgehammer, once one of the external provers found a proof, it is likely that it can be reproduced inside Isabelle (but not always, as reported by Paulson and Blanchette (2012)). The external provers significantly reduce the number of premises passed to the reproduction step, therefore the Isabelle's proof will be short. The major bottleneck of Sledgehammer, however, is the pre-selection step: the external provers often cannot find a proof because they are provided too few – or too many – premises.

In Magnushammer, on the other hand, we skip the external provers completely and input premises directly into the native Isabelle's tactics to produce a proof. This means that the prediction must be of high quality in order to obtain good results. The number of the premises will be typically larger – therefore the proofs will be longer, and of form of a combination of a strong tactic and a long list of premises as its arguments.

As an example demonstrating the difference between Magnushammer and Sledgehammer from the perspective of produced proofs, let's see two proofs of the algebraic theorem `set_r_ar_cos_ker` from the Archive of Formal Proofs:[3]

Sledgehammer's proof:

```
by (smt (z3) Ring.set_r_ar_cos ker_ideal)
```

Magnushammer's proof:

```
by (clarsimp simp add: set_ar_cos_def Ring.Ring Ring.set_r_ar_cos
eq_prop Ring.I_in_set_ar_cos Set.bexE Ring.ring_is_ag ker_ideal)
```

Both Sledgehammer and Magnushammer were able to solve it, however, the latter used more premises. This is expected: whenever both methods find a proof, the Magnushammer's proof is often longer in the sense of the number of premises used. Yet, Sledgehammer's weaker pre-selection scheme causes it to find fewer proofs in comparison.

An example of a theorem that Sledgehammer was unable to prove (with a generous time limit of 60 s), but Magnushammer has proven, is lemma `unit_disc_fix_moebius_uminus`.[4] The proof

---

[3]from the theory Group-Ring-Module/Algebra4.thy, accesible at `https://search.isabelle.in.tum.de/#theory/default_Isabelle2022_AFP2022/Group-Ring-Module/Algebra4`

[4]from the theory Complex_Geometry/Unit_Circle_Preserving_Moebius.thy, accessible at `https://search.isabelle.in.tum.de/#theory/default_Isabelle2022_AFP2022/Complex_Geometry/Unit_Circle_Preserving_Moebius`

produced by Magnushammer consists of the smt tactic and a list of premises. Thus, Magnushammer was able to retrieve the necessary premises in contrast to Sledgehammer:

```
by (smt (z3) unit_disc_fix_unit_circle_fix
Oriented_Circlines.unit_disc_def unit_circle_fix_moebius_uminus
unit_disc_fix_moebius_comp Set.image_iff unit_disc_fix_iff
Moebius.uminus_moebius_def Unitary11_Matrices.unitary11_unitary11_gen
unit_disc_fix.abs_eq Oriented_Circlines.inf_notin_unit_disc
Moebius.plus_moebius_def unit_disc_fix_discI unit_disc_fix_moebius_add
unit_disc_fix_id_moebius Set.imageE Set.imageI
Oriented_Circlines.zero_in_unit_disc SMT.verit_minus_simplify(4)
unit_circle_fix_moebius_comp
```

