# OpenReview forum: "Magnushammer: A Transformer-Based Approach to Premise Selection"
_ICLR.cc/2024/Conference — ICLR 2024 poster_

### Official Review · Reviewer_XHfH · 2023-10-13

**Soundness:** 3 good
**Presentation:** 3 good
**Contribution:** 3 good
**Rating:** 8
**Confidence:** 4

**Summary:**

This paper proposes a new method for premise selection in automated theorem proving. The proposed method is based on training a transformer model on the text of the lemmas that are available in a library. The proposed method improves the testing accuracy of AI assistants on some benchmarks.

**Strengths:**

The open source premise selection dataset appears to me as a useful contribution for the community.

I found the approach and the proposed method interesting.

Paper is well structured and its writing is clear.

**Weaknesses:**

The main shortcoming that appears to me is the lack of comparison with recent methods such as LeanDojo. Perhaps authors can address this during the rebuttal, clarifying the advantage of their proposed method.

Specifically, in the list of novel contributions of the paper, the first item is stated to be the use of contrastive learning for premise selection. However, contrastive learning was previously used in LeanDojo. It would be helpful if authors clarify the specifics of their contribution compared to previous methods.

-----

Moreover, authors have not cited the reference below which also trains a GPT model on mathematical statements. It seems that the authors' use of the GPT-2 model has some similarities with the approach taken by the paper below. So, it is not clear to me whether authors' approach is as novel as it is claimed to be. I ask that authors clarify their contributions in relation to this reference as well.

[1] Polu, S., Han, J.M., Zheng, K., Baksys, M., Babuschkin, I. and Sutskever, I., 2022, September. Formal Mathematics Statement Curriculum Learning. In The Eleventh International Conference on Learning Representations.


------------
As a limitation of Sledgehammer, this paper mentions the need for handcrafted design. In contrast to Sledgehammer, it is mentioned that Magnushammer does not need handcrafted design and feature engineering. However, the choice of negative premises appeared to me as a handcrafted approach in Magnushammer. I do not think this is necessarily a shortcoming of the Magnushammer, but I think the benefits and contributions of the Magnushammer might have been magnified a bit.

**Questions:**

Would it be possible for the authors to expand on their improvement on the miniF2F benchmark? Specifically, I would be interested to know what are the specific unsolved theorems that Magnushammer was able to solve, and how complex are those proofs. When one tries to solve those theorems with Sledgehammer or other assistants, what exactly goes wrong, and what is the gap that Magnushammer can bridge while sledgehammer cannot? Are the premises that Magnushammer can find, and Sledgehammer fails to find, related to the premises that were chosen by the authors for their contrastive learning? I think giving maybe two or three examples (theorems from miniF2F that were solved by Magnushammer) would improve the presentation of this paper’s contribution.

Authors mention that one benefit of Magnushammer is that it can be trained on various languages, and it is not limited to Isabelle, etc. Have authors experimented with lean? I did not see any results on lean, but I might have missed it.

---

> ### Author Response · Authors · 2023-11-18
>
> 1. *How is your work different from LeanDojo?*
>
> LeanDojo [1] indeed shares similarities with our work: it also tackles the problem of automating proof search in proof assistants using neural language models. Moreover, it also uses contrastive learning to train a premise selection model. Still, there are important differences between the two works: (1) a major architectural difference is that in our method there is an additional Rerank phase that refines the prediction from the Select phase allowing the tokens of the proof state directly attend to tokens of the premise, giving a more contextualized relevance score. LeanDojo, on the other hand, retrieves premises using the approach of the Select phase only. (2) LeanDojo is designed for Lean, and it develops a Lean-specific toolkit, whereas we experimented with Isabelle. This implies different evaluation benchmarks: LeanDojo uses ProofNet and miniF2F, and we use PISA and miniF2F (as the latter is a multi-ITP benchmark). (3) There are also a few other, less prominent, differences, e.g., LeanDojo applies a bit different scheme of sampling negatives for contrastive training.
>
> Our work was completed before LeanDojo's release, and the two works should be treated as concurrent. (To comply with the anonymity rules we do not present evidence for this claim. We would be happy to do this if instructed so by AC.) The similarities as encountered between LeanDojo and our work are often typical in scientific research when subsequent studies build upon existing work.
>
> 2. *How is your work different from the work of Polu et al. [2]?*
>
> Polu et al. [2], similarly to our work, study the problem of automating theorem proving in proof assistants using neural language models. Their work, however, differs in several aspects: (1) First of all, they focus on neurally-guided proof search, whereas our work tackles premise selection (a challenging aspect of proof search). (2) They train models using different objectives: proof step objective (predicting the next proof step given a proof state) or proof size objective (predicting the length of the proof closing the current goal) that given a proof state predicts the next proof step. We train a model using contrastive objective on (proof state, premise) pairs, as well as Rerank model. (3) They explore the potential of expert iteration technique, where the proof searches and training are interspersed and run on a curriculum of increasingly difficult problems (this is orthogonal to our approach). (4) They evaluate their approach in Lean, whereas we use Isabelle. This implies several differences: Isabelle’s library is more mature and larger, giving more training data. Lean lacks strong tactics able to perform automated search, whereas Isabelle has many of them (like metis, smt, etc.); the bottleneck of these tactics is precisely premise selection. In Lean, the difficulty of automated proving is placed a bit differently: it lies not only in premise selection, but to a larger extent also in combining weaker tactics together.
>
> 3. *Isn’t your approach to selecting negative premises a handcrafted element in the architecture of Magnushammer?*
>
> First, to clarify the notions, by a handcrafted element of the premise selection architecture we understand these operations in the pipeline that need to be implemented employing special domain knowledge. In Sledgehammer one can identify at least two instances of such handcrafted operations: (1) In the pre-filtering schemes for selecting premises to be passed to the external provers, where either heuristics are employed or classical machine learning models relying on pre-designed features; these are based on the syntactic structure of the logical expressions. (2) As the input to the external provers is based on different logics than that of Isabelle, it enforces a nuanced translation step. Both (1) and (2) require manual implementations presupposing thorough understanding of the logic and technical details of Isabelle or the external provers.
>
> Now, regarding the selection of the negatives for training: once a dataset of positive examples is collected, it happens entirely automatically: in the Select stage, the negatives are simply sampled from all the available non-positive premises, and in the Rerank stage, the negatives deemed positive by the Select model are used as (hard) negative examples for training the Rerank model whose task is to refine the predicted ranking. There is no domain specific knowledge involved in the process.
>
> Another way to think about it is that the selection of negative premises is a step to create a training dataset which could, potentially be replaced with other approaches, and does not take part in the process of premise selection during inference -- the goal is to have an autonomous agent reasoning over premises and making decisions. This is in contrast to Sledgehammer which uses its heuristics explicitly during inference and thus its capabilities are inherently limited by them.

---

> ### Author Response · Authors · 2023-11-18
>
> 4. *What are the specific unsolved theorems that Magnushammer was able to solve? How complex are those proofs? What are the shortcomings of Sledgehammer that Magnushammer overcomes? Please, give a few examples.*
>
> The main shortcoming of Sledgehammer that Magnushammer addresses is the ability to perform high-quality premise selection without the complicated, highly engineered pipeline involving pre-selection of premises, translation to different logic, running external provers, and finally reproducing the proof inside Isabelle. Magnushammer demonstrated that it is possible to provide premise selection model strong enough that we can part with the external provers altogether and obtain better results. Magnushammer operates purely on textual representations of data, which is easier to create and reason about, whereas Sledgehammer requires complex typing and engineering.
>
> In Sledgehammer, once one of the external provers found a proof, it is likely that it can be reproduced inside Isabelle (but not always!). The external provers significantly reduce the number of premises passed to the reproduction step, therefore the Isabelle’s proof will be short. The major bottleneck of Sledgehammer, however, is the pre-selection step: the external provers often cannot find a proof because they are provided too few – or too many – premises.
>
> In Magnushammer, on the other hand, we skip the external provers completely and input premises directly into the native Isabelle’s tactics to produce a proof. This means that the prediction must be of high quality in order to obtain good results. The number of the premises will be typically larger – therefore the proofs will be longer, and of form of a combination of a strong tactic and a long list of premises as its arguments.
>
> As an example demonstrating the difference between Magnushammer and Sledgehammer from the perspective of produced proofs, let’s see two proofs of the algebraic theorem `set_r_ar_cos_ker` from the Archive of Formal Proofs (from theory Group-Ring-Module/Algebra4.thy [3]):
>
> Sledgehammer’s proof:
> ```
> by (smt (z3) Ring.set_r_ar_cos ker_ideal)
> ```
> Magnushammer’s proof:
> ```
> by (clarsimp simp add: set_ar_cos_def Ring.Ring Ring.set_r_ar_cos eq_prop Ring.I_in_set_ar_cos Set.bexE Ring.ring_is_ag ker_ideal)
> ```
> Both Sledgehammer and Magnushammer were able to solve it, however, the latter used more premises. This is expected: whenever both methods find a proof, the Magnushammer’s proof is often longer in the sense of the number of premises used. Yet, Sledgehammer’s weaker pre-selection scheme causes it to find fewer proofs in comparison.
>
> An example of a theorem that Sledgehammer was unable to prove(with a generous time limit of 60 s), but Magnushammer has proven, is lemma `unit_disc_fix_moebius_uminus` (from the theory Complex_Geometry/Unit_Circle_Preserving_Moebius.thy [4]). The proof produced by Magnushammer consists of the smt tactic and a list of premises. Thus, Magnushammer was able to retrieve the necessary premises in contrast to Sledgehammer:
> ```
> by (smt (z3) unit_disc_fix_unit_circle_fix Oriented_Circlines.unit_disc_def unit_circle_fix_moebius_uminus unit_disc_fix_moebius_comp Set.image_iff unit_disc_fix_iff Moebius.uminus_moebius_def Unitary11_Matrices.unitary11_unitary11_gen unit_disc_fix.abs_eq Oriented_Circlines.inf_notin_unit_disc Moebius.plus_moebius_def unit_disc_fix_discI unit_disc_fix_moebius_add unit_disc_fix_id_moebius Set.imageE Set.imageI Oriented_Circlines.zero_in_unit_disc unit_disc_fix.rep_eq unit_disc_fix_def unit_disc_fix_iff_ounit_circle SMT.verit_minus_simplify(4) unit_circle_fix_moebius_comp
> ```
>
> 5. *Have authors experimented with Lean?*
>
> Taking advantage of the presence of LeanDojo, the interface with Lean it implements, and the neural prover associated with it – ReProver [5] – we have started experiments aimed at substituting premise selection of ReProver with Magnushammer to see if an improvement can be achieved on miniF2F and ProofNet benchmarks. We are currently training a Magnushammer model on Lean data obtained via LeanDojo. We also plan to see the effect of training jointly on Lean and Isabelle data. Once we obtain the results, we will report them in the appendix of the camera-ready version of the paper (or, being optimistic, already in this discussion.)
>
> **References**
>
> [1] Yang et al., LeanDojo: Theorem Proving with Retrieval-Augmented Language Models. arXiv 2023
>
> [2] Polu et al., Formal Mathematics Statement Curriculum Learning. ICLR 2022
>
> [3] https://search.isabelle.in.tum.de/#theory/default_Isabelle2022_AFP2022/Group-Ring-Module/Algebra4
>
> [4] https://search.isabelle.in.tum.de/#theory/default_Isabelle2022_AFP2022/Complex_Geometry/Unit_Circle_Preserving_Moebius
>
> [5] https://github.com/lean-dojo/ReProver

---

> > ### Comment · Reviewer_XHfH · 2023-11-20
> >
> > I thank the authors for their clear and thoughtful response. The provided examples and clarifications address my questions -- I adjust my score.
> >
> > I suggest authors include these (and possibly more) examples in their paper as well as the discussions that clarify their contributions.

---

> > > ### Author Response · Authors · 2023-11-21
> > >
> > > We are really grateful to the reviewer for appreciating our response. As suggested, we included the provided examples and some of the explanations in Appendix F. Moreover, we emphasized the difference between LeanDojo and Magnushammer in Section 6, and we added a reference to Polu et al. [2] there.

---

### Official Review · Reviewer_7Qsu · 2023-10-31

**Soundness:** 4 excellent
**Presentation:** 4 excellent
**Contribution:** 2 fair
**Rating:** 8
**Confidence:** 5

**Summary:**

This paper Magnushammer, a transformer model trained for premise selection on Isabelle. They first build a training set of premise selection from human-written proofs and machine-generated proofs produced by Sledgehammer. Magnushammer select relevant premises in two steps, first select a larger set of premises from all candidate based on cosine similarity of embeddings from a transformer model, then rerank these the selected premises by concating them with the proof state and feeding the new sequences into the transformer for binary classification. Experiments validate that, compared to Sledgehammer, better premise selection using Magnushammer leads to significant improvement for theorem proving with Thor on the PISA (57% -> 71%) and miniF2F (29.9% -> 37.3%) dataset.

**Strengths:**

1 Similar to LeanDojo, this paper demonstrates that better premise selection is crucial for the success of theorem proving.
2 The proposed approach is technically sound, with properly designed select and rerank steps, InfoNEC and BCE losses and data collection steps.
3 Experiments demonstrate significant improvement resulting from Magnushammer.
4 The proposed dataset for premise selection could facilitate the future progress for automated theorem proving.

**Weaknesses:**

Still, the main story of this paper is almost the same as LeanDojo. The proposed two-step pipeline and training method are also commonly used.

**Questions:**

1 It is great to know we could advance theorem proving by developing better premise selection approaches. My question is what the upper bound is given our current recipe. Is it possible to obtain the results of Thor with ground truth premise selection?
2 Did you retrain Thor with Magnushammer or just apply the original Thor trained with Sledgehammer to Magnushammer?
3 Could you quantitatively evaluate the contribution of Magnushammer for theorem proving with Thor, such as how many proof steps call Magnushammer in your test.

**Details Of Ethics Concerns:**

No ethics concerns.

---

> ### Author Response · Authors · 2023-11-18
>
> 1. *What is the upper bound given our current recipe. Is it possible to obtain the results of Thor with ground truth premise selection?*
>
> Having an upper-bound would be indeed useful, however, there is no easy way to achieve it. The reason for this is that the proof steps generated with Thor often differ from the ground truth steps from the proof corpus, for which we do not have any “ground truth premises”.
>
> 2. *Did you retrain Thor with Magnushammer or just apply the original Thor trained with Sledgehammer to Magnushammer?*
>
> We retrain Thor with the original recipe. To clarify further, the training process of Thor does not involve Magnushammer at all: we simply use heuristics to find the steps in the proof corpus that were produced by Sledgehammer and replace them with an identifier sequence. How to improve on this recipe to make it Magnushammer-aware is an interesting research question that we leave for future work.
>
> 3. *Could you quantitatively evaluate the contribution of Magnushammer for theorem proving with Thor, such as how many proof steps call Magnushammer in your test?*
>
> Thank you for this suggestion. Unfortunately, we have not logged such information. We will run our somewhat costly evaluations and provide numbers in the camera-ready version.
>
> 4. *The main story of this paper is almost the same as LeanDojo. The proposed two-step pipeline and training method are also commonly used.*
>
> This is indeed a justifiable observation. However, note that our work was completed before LeanDojo’s release, and the two works should be treated as concurrent. Such similarities are often typical in scientific research when subsequent studies build upon existing work. ​​We showed how to effectively combine existing methods to achieve a generic premise selection tool that is a substantial improvement over Sledgehammer both in single- and multi-step settings (one-step proofs and Thor, respectively).
> Also, please, see our answer to the question 1 of the reviewer XHfH where we say a bit about the differences between LeanDojo and our work.

---

### Official Review · Reviewer_uz6C · 2023-10-31

**Soundness:** 3 good
**Presentation:** 3 good
**Contribution:** 3 good
**Rating:** 8
**Confidence:** 3

**Summary:**

In the field of automated theorem proving, premise selection is the process of selecting a subset of mathematical statements that are already known to be true as a set of premises to build new knowledge. In this work, the authors introduce MAGNUSHAMMER, a premise selection technique based on transformers.

**Strengths:**

The idea of using transformers in for premise selection is quite interesting. In my opinion, the main strength of this paper the fact that this approach was able to outperform an automated theorem prover based on similar paradigm by a good margin. More specifically, the new approach obtained a proof rate of 59.5%, while sledgehammer, which is one of the most well known premise selection tools for Isabelle, obtained a proof rate of 38.3%.

The authors also contribute with a large dataset for benchmark of premise selection paradigms.

Finally, the authors provide a careful analysis of their approach by analysing the dependence on quantities such as model size, dataset size. According to their results, when resources are moderate, their approach outperforms existing techniques.

**Weaknesses:**

The main drawback seems to be the fact that the tools were evaluated only on two datasets (PISA and miniF2F).

**Questions:**

What are other suitable datasets for the evaluation of tools for premise selection?

---

> ### Author Response · Authors · 2023-11-18
>
> 1. *What are other suitable datasets for the evaluation of tools for premise selection? The main drawback of the paper seems to be the fact that the tools were evaluated only on two datasets (PISA and miniF2F).*
>
> In general, the premise selection problem appears both in the context of automated and interactive theorem proving. There are multiple provers in both categories, and they are typically based on different, often incompatible logics – therefore each of them is associated with different sets of benchmarks. For instance, one of the more prominent datasets for evaluating premise selection in automated theorem proving is this used in [1], which is derived from Mizar Mathematical Library, translated to TPTP [2] – the standard formalism used by automated theorem provers. On the other hand, ProofNet [3] is a recently released benchmark specific for the Lean interactive theorem prover that supports evaluation of several tasks, including automated proof search and premise selection.
>
> In this work, we focused on premise selection for interactive theorem proving, and we decided to use the Isabelle’s ecosystem to develop and evaluate our approach. To the best of our knowledge, PISA and miniF2F benchmarks are the only published Isabelle’s datasets previously used to evaluate machine learning automation for Isabelle, thus we decided to evaluate our approach on them.
>
> **References**
>
> [1] Irving et al., DeepMath -- Deep Sequence Models for Premise Selection. NIPS 2016
>
> [2] Sutcliffe: The TPTP World -- Infrastructure for Automated Reasoning. LPAR 2010
>
> [3] Azerbayev et al., ProofNet: Autoformalizing and Formally Proving Undergraduate-Level Mathematics. arXiv 2023

---

### Official Review · Reviewer_t7yc · 2023-11-01

**Soundness:** 4 excellent
**Presentation:** 4 excellent
**Contribution:** 4 excellent
**Rating:** 8
**Confidence:** 4

**Summary:**

The paper introduces a language-model-based premise selection tool, Magnushammer, for the Proof assistant Isabelle. Given the current textual proof state, Magnushammer, is able to select a set of relevant premises based on trained transformer models. This selection is two-staged: (1) SELECT: an efficient step that selects premises with the highest representation similarity to the given proof state; (2) RERANK: re-rank the selected premises based on a score outputted by a binary classifier predicting whether the proof state and the premise are relevant. The models in both stages are trained contrastively, upon the data collected from human-proof libraries and alternative proofs from Sledgehammer. During experiments, Magnushammer shows significant improvement over Sledgehammer. The authors also open-sourced the collected premise-selection dataset.

**Strengths:**

The paper proposes a novel approach to premise selection, which was previously considered a difficult task for LLMs since the length of relevant premises is usually much larger than what LLMs can handle. With batch-contrastive learning and careful data design and collection, Magnushammer retrieves premises instead of generating them using Transformers, showing impressive performance on theorem-proving benchmarks.

•	The experiment design is generally sound and compelling. The experiment separates the training and test data and evaluates the quality of premise selection in both single-step (direct evaluation) and multi-step (integration with Thor) tasks. The authors also present the impact of the number of selected premises on the proof rate, as an alternative measurement of the premise selection quality.

•	The ablation study is comprehensive, covering the pre-trained model, model size, dataset size, and compute budget.

**Weaknesses:**

The presentation of why the premise selection was considered difficult for LLMs, and how the proposed method tackles these difficulties is somehow weak. Some more examples and explanations/discussion on LLM/contrastive learning would help.

•	Section 6 mentions GPT-f, PACT, Thor, and other LLM-based theorem-proving efforts, highlighting their diverse focuses. It would be nice if the authors provided a more insightful and coherent discussion on how different aspects are connected (e.g., premise selection, tactic prediction, proof step search), how the previous works explicitly or implicitly tackled the premise selection problem, and how the proposed method is advancing the area.

•	Section 3 Training part mentions that “This gives N-1+M negatives per proof state in one batch”. Is it possible that a positive premise for one state is also positive for another state in the same batch? Did you select positive examples intentionally avoiding this possibility, or, in this case, the negative examples are less than N-1+M?

•	During the evaluation, for each proof state, Magnushammer parallelly checks whether a proof can be completed from premise subsets of different sizes. The author claims that similar techniques are also implemented in Sledgehammer. The author can explain more on this point to ensure the experimental comparison between Magnushammer and Sledgehammer is fair, e.g., in terms of parallelism, number of selected subsets, and their sizes.

•	Minor formatting issue: the reference of Thor in Section D.2 on Pg21.

**Questions:**

•	The author mentioned that the MAPL dataset “decreases the probability of sampling false negatives while training contrastively”. Could the author provide further details on the likelihood of a premise being positive if it is not found by human-proof collections and Sledgehammer? Additionally, are there other methods to validate or enhance the confidence that it is a true negative? Also, the influence of false negatives on batch-contrastive learning can be elaborated.

---

> ### Author Response · Authors · 2023-11-18
>
> 1. *The MAPL dataset “decreases the probability of sampling false negatives while training contrastively”. What is the likelihood of a premise being positive if it is not found by human-proof collections and Sledgehammer?*
>
> This is an interesting and quite nuanced question. From the perspective of proof calculus, any premise may be seen as “positive", as it can be used in every proof, possibly in a “redundant” way. One could also argue that a premise is positive if it is contained in one of the minimal sets of premises appearing in the valid proofs. In practice, the automated methods often are unable to find a proof depending on such a minimal set of premises and some premises may not end up in a proof but still influence the automated search in a positive direction. Consequently, we can only think about an approximate notion of positive and negative premises, where positive means that including it increases the chances of automatically finding a proof, whereas negative means the opposite. Having said that, the number of new premises captured as positives (i.e., such that would previously be counted as negatives) on MAPL for theorems proved both by a human and Sledgehammer average to 9.85.
>
> 2. *Are there other methods to validate or enhance the confidence that a premise is a true negative?*
>
> The only method (we can think of) of eliminating false negatives in premise selection is via repetitively running an automated prover on different sets of premises and collecting as many different proofs as possible. Doing this exhaustively is practically infeasible, so one strategy is to run the prover against sets of premises pre-selected by a data-driven / heuristical method, and by this exploring premises that are more likely to be positive.This strategy seems to be the best, nevertheless, it is still imperfect: some positive premises will not be captured by the pre-selection method, and therefore will constitute false positives.
>
> 3. *Please, comment on the influence of false negatives on batch-contrastive learning.*
>
> In most practical scenarios (not only in premise selection) false negatives are unavoidable in batch-contrastive learning, especially in unsupervised settings. In practice, the presence of this issue does not prevent contrastive learning approaches from achieving good, often state-of-the-art, results on a range of different tasks [1,2]. Having said that, there are works that measure the negative influence of false negatives in contrastive learning [3,4]. This problem becomes especially pronounced when one applies hard negative mining in contrastive learning, i.e., sampling negatives that appear to be similar for the model. Hard negative mining is beneficial but also increases the probability of false negatives, which creates a trade-off (which [4] addresses).
>
> 4. *Is it possible that a positive premise for one state is also positive for another state in the same batch? Did you select positive examples intentionally avoiding this possibility, or, in this case, the negative examples are less than N-1+M?*
>
> That is a good question from the reviewer. In this case, such a premise is treated as a negative from the loss calculation point of view. However, it might be a false negative as in the scenario described by the reviewer. We didn't address this issue for simplicity and the fact that false negatives are omnipresent in the premise selection dataset. But we agree that it could be improved in our training pipeline. Nevertheless, despite this imperfect labeling, contrastive learning allows us to train models.
>
> 5. *Please, compare the strategies of trying different subsets of premises by Magnushammer vs. Sledgehammer.*
>
> Magnushammer tries multiple subsets of premises as inputs to Isabelle’s tactics. In contrast, Sledgehammer inputs different subsets of premises as inputs to external provers, whereas the input of tactics consists of all the premises judged useful by the provers. The strategy of selecting the subsets of premises by Sledgehammer is complicated and not easily comparable to the strategy of Magnushammer. For instance, “the number of facts included in a problem varies from prover to prover, since some provers get overwhelmed more easily than others” [5].

---

> ### Author Response · Authors · 2023-11-18
>
> 6. *It would be nice if Section 6 provided more insightful characterization of the landscape of LLM-based theorem provers and how Magnushammer is positioned there. The explanation of the difficulty of premise selection by LLMs and how Mgnushammer overcomes them could be improved.*
>
> We definitely agree that the field of “neural theorem proving” has become rich and diverse enough that it deserves more deep discussion, characterization, and comparative analysis. A special emphasis should be put on the premise selection problem, which is inherent to all the large corpora of formalized knowledge. The format of ICLR submission will not allow for a thorough treatment of the topic, but we are going to update the paper to better highlight the contributions of Magnushammer by contrasting it with different approaches. (Extending the paper turned out not entirely easy as the 9 pages are already filled compactly, but we are doing our best and will upload an updated version soon). Thank you, these were very meaningful suggestions!
>
> **References**
>
> [1] Neelakantan et al., Text and Code Embeddings by Contrastive Pre-Training. arXiv 2022
>
> [2] Izacard et al., Unsupervised Dense Information Retrieval with Contrastive Learning. TMLR 2022
>
> [3] Chuang et al., Debiased Contrastive Learning. NeurIPS 2020
>
> [4] Robinson et al., Contrastive Learning with Hard Negative Samples. ICLR 2021
>
> [5] https://isabelle.in.tum.de/dist/doc/sledgehammer.pdf, page 7

---

> > ### Author Response · Authors · 2023-11-21
> >
> > As suggested by the reviewer, in the revised submission we extended and slightly reorganized Section 6 (Related work) in attempt to provide more coherent and insightful story surrounding the problem of premise selection, how other works tackle it, and how Magnushammer contrasts with them. We would be grateful for a feedback if the improved text sufficiently addresses the remarks of the reviewer. In order to comply with the space limitations, we removed the last short section (Conclusion) which mostly consisted of reiterating what was said elsewhere and brought less value than the extended picture of the related work. For the same reason, we also moved a paragraph concerning the ablation for re-ranking to Appendix C.5.

---

### Author Response · Authors · 2023-11-18

We thank the reviewers for taking the time to review our work and sharing their thoughtful comments, which gave us an opportunity to further improve our work.

Overall, we are thrilled that the reviewers were in broad agreement on the following important points:
* Our transformer-based approach to premise selection represents good progress in the field of automated reasoning.
* The experiment design is sound and compelling.
* The analysis of the results is careful and thorough.
* The new open-source dataset for premise selection constitutes a useful contribution to the field.

We address specific questions in the responses to individual reviews.

---

### Author Response · Authors · 2023-11-23
**Summary of changes in the revised version of the submission**

We wanted to again sincerely thank the reviewers for their work and thoughtful feedback. Below, we summarize the changes we introduced to the revised version of our submission.

* In response to the reviewer t7yc, we extended and reorganized Section 6 (Related work) in attempt to provide more coherent and insightful story surrounding the problem of premise selection, how other works tackle it, and how Magnushammer contrasts with them. As this took some space, in order to comply with the 9-page limit, we removed the last short section (Conclusion) which mostly consisted of reiterating what was said elsewhere and brought less value than the extended picture of the related work. For the same reason, we also moved a paragraph concerning the ablation for re-ranking to Appendix C.5.
* In response to the reviewer XHfH, in Appendix F we included some examples of proofs produced by Magnushammer and compared them to proofs produced by Sledgehammer.
* We added a reference to Polu et al. [1] in Section 6.
* We made some minor formatting fixes, like a reference to Thor in Appendix D.2, pointed out by the reviewer t7yc.

[1] Polu et al., Formal Mathematics Statement Curriculum Learning. ICLR 2022

---

### Meta-Review · Area_Chair_YZEz · 2023-12-08

**Metareview:**

This work addressed premise selection for theorem proving. It receives unanimous positive reviews and rebuttal has addressed some concerns about the novelty. The method of premise selection leads to significant performance gain for theorem proving, and the dataset released is valuable for the community. AC agrees this is a nice work for an important new area of AI like automated theorem proving, but also shares the same concerns on its similarity to LeanDojo, and thus recommends acceptance as a poster.

**Justification For Why Not Higher Score:**

The framework is either quite similar to related works, or otherwise commomly used.

**Justification For Why Not Lower Score:**

The method is technically sound with good performance. The dataset open-sourced is the largest available premise selection dataset, and the first one for Isabelle.

---

### Decision · Program_Chairs · 2024-01-16

Accept (poster)